# The Ecophysiological Performance and Traits of Genera within the *Stichococcus*-like Clade (Trebouxiophyceae) under Matric and Osmotic Stress

**DOI:** 10.3390/microorganisms9091816

**Published:** 2021-08-26

**Authors:** Anh Tu Van, Veronika Sommer, Karin Glaser

**Affiliations:** Institute for Biological Sciences, Applied Ecology and Phycology, University of Rostock, 18059 Rostock, Germany; veronika.sommer@uni-rostock.de (V.S.); karin.glaser@uni-rostock.de (K.G.)

**Keywords:** *Stichococcus*, *Pseudostichococcus*, osmotic stress, desiccation tolerance, osmolytes

## Abstract

Changes in water balance are some of the most critical challenges that aeroterrestrial algae face. They have a wide variety of mechanisms to protect against osmotic stress, including, but not limited to, downregulating photosynthesis, the production of compatible solutes, spore and akinete formation, biofilms, as well as triggering structural cellular changes. In comparison, algae living in saline environments must cope with ionic stress, which has similar effects on the physiology as desiccation in addition to sodium and chloride ion toxicity. These environmental challenges define ecological niches for both specialist and generalist algae. One alga known to be aeroterrestrial and euryhaline is *Stichococcus bacillaris* Nägeli, possessing the ability to withstand both matric and osmotic stresses, which may contribute to wide distribution worldwide. Following taxonomic revision of *Stichococcus* into seven lineages, we here examined their physiological responses to osmotic and matric stress through a salt growth challenge and desiccation experiment. The results demonstrate that innate compatible solute production capacity under salt stress and desiccation tolerance are independent of one another, and that salt tolerance is more variable than desiccation tolerance in the *Stichococcus*-like genera. Furthermore, algae within this group likely occupy similar ecological niches, with the exception of *Pseudostichococcus*.

## 1. Introduction

Terrestrial algae, abundant in both extreme and temperate environments, are exposed to several abiotic stressors related to their habitat outside the aquatic environment such as UV radiation, freezing, and unpredictable water availability, where periods of drought of several weeks or even months punctuated by rains are common [1,2,3]. The subsequent success of terrestrial algae is partly due to their ability to withstand the stress of undergoing repeated and extended drying–rewetting cycles [4]. Many terrestrial green algae have a palette of drought tolerance through adaption mechanisms, such as the evolution of flexible cell walls, an evaporation barrier formed by the excretion of extracellular polysaccharides (EPS) [5,6,7] and physiological adaption by the production of osmoprotectant compounds [8,9,10].

Matric stress on algae is caused unequal water attraction potentials (Ψ_m_) between the cell and environment [11]. In this study, it is analogous to desiccation (dehydration to below 50% relative humidity (RH) [1]), although matric stress can be induced by using substances that can induce Ψ_m_ differences, e.g., polyethylene glycol 8000 [12]. In addition to desiccation, osmotic stress in the form of salt in the surrounding environment may be significant, e.g., in marine environments. While dry conditions reduce the amount of water available from the environment, osmotic stress disrupts the normal cellular ion concentrations; furthermore, Na^+^ and Cl^−^ ions are toxic in high concentrations [13]. Two major physiological responses to desiccation and osmotic stress are the reduction in photosystem II (PSII) activity and the production of low-molecular-weight carbohydrates (LMWC) as compatible solutes, also known as osmolytes [1]. LMWC concentrations, particularly sugar alcohols and simple sugars, have been shown to increase under salt stress conditions in various algal groups (Trebouxiophyceae [8]; *Heterococcus* [14]; *Chlorella* [15]; various seaweeds [16]). A recent transcriptomic analysis [17] had shown that under salt and nitrogen stress, the chlorophyte alga *Neochloris oleoabundans* upregulates genes coding for osmoregulation, particularly those for compatible solutes.

Various taxa within the *Stichococcus*-like genera of the Prasiolaceae (Trebouxiophyceae) are ubiquitous and abundant in diverse terrestrial habitats, such as tree bark [18,19], surfaces of buildings [20,21,22], biocrusts [2], and soil [23]. Due to their simple morphology and resemblance to one another, they have long been functionally grouped into the *Stichococcus* clade. This group currently includes *Stichococcus*, *Protostichococcus*, *Deuterostichococcus*, *Tritostichococcus*, *Tetratostichococcus*, *Desmococcus* as well as *Diplosphaera* [24]. Prior research shows that *Stichococcus bacillaris* can occur, albeit rarely, in marine conditions [25] and is very halotolerant [8,26,27]. Although *Stichococcus* are known to produce sorbitol, sucrose, and proline as osmoprotectants [28,29], there lacks comparative examinations of its response to osmotic as well as matric stress; this is also true for the other genera within the *Stichococcus*-like clade. Here, we aim to characterize the ecophysiological performance of *Stichococcus*-like strains to these pragmatically realized tolerance to salt and desiccation.

## 2. Materials and Methods

Twelve established unialgal strains from around the world were used (Table 1). All cultures were maintained at 20–22 °C on 1.5% modified Bold’s Basal agar (3N BBM+V [30], modified to have triple nitrate concentration [31]) with a 16:8 light: dark photoperiod, with 30 µm photons m^−2^ s^−1^ (Lumilux Cool Daylight L18W/840, OSRAM, Munich, Germany).

### 2.1. Salt Tolerance

The halotolerance of select strains (Table 1) was determined by exposing them to salinity along a gradient corresponding to natural freshwater, sea water as well as to extreme experimental conditions. These strains were chosen on their ability to produce significant biomass in a short period. A series of nine media was prepared, with salinities ranging from 0 to 90 g NaCl L^−1^ in 15 g L^−1^ increments, corresponding to their values in absolute salinity (S_A_). An amount of 30 mL medium was pipetted into standard 150 mm glass test tubes, so that one salinity level corresponded to one test tube. Inoculation with 1 mL unialgal culture suspensions in the respective salinities followed. The cultures grew for eight weeks; the tubes were photographed weekly and aliquots from the test tubes periodically observed under the microscope for morphological changes. After the experiment, growth was classified by eye using the following categories: healthy, slightly stressed, stressed, mostly dead, and dead.

### 2.2. Desiccation Tolerance and Recovery Experiments

The experimental setup for desiccation tolerance and recovery largely followed that of Karsten [32] with minor changes to the setup and measurement protocol outlined below. Strains were kept in the exponential growth phase by transfer every three days before experiment began. During the experiment, medium was refreshed daily through decanting and replacement of 25 mL of old medium with fresh, following biomass collection.

For the dehydration experiment, polypropylene desiccation chambers were filled with 100 g freshly activated silica; a perforated metal disc was positioned upon four columns within the chambers. Six Whatman GF/F 25 mm fiberglass filters were distributed along the edge of the metal disk, and 200 µL suspensions of each strain were dropped onto the filters, so that one filter corresponded to one strain. A data logger monitored air humidity and temperature, which were maintained at RH of ~10% and 20–22 °C, respectively. Throughout the duration of the experiment, all polystyrene chambers and algae were kept in a low-light environment (ca. 40 µm photons m^−2^ s^−1^). The effective quantum yield of PSII (Y(II), ΔF/Fm’) of each strain was measured every 30 min until no yield was recorded (420 min) on a PAM 2500 (Walz, Effeltrich, Germany).

The rehydration experiment following the dehydration period was set up in a similar way, except that the silica gel was replaced with 100 mL tap water in a new chamber to create a moist atmosphere (RH ~95%). The filters were moved to the high-humidity chambers and rehydrated with 200 µL of algae growth medium, after which the effective quantum yield was measured every 30 min up to 2 h; a final measurement was done after 24 h of recovery. The strain *Desmococcus olivaceus* SAG 1.92 was not tested due to insufficient accumulation of biomass before experiment begin.

Three technical experimental replicates and two control replicates were taken over a period of one week, where one replicate run took one day. The control replicates were tested at the beginning and end of the experiment, to account for changes in culture vitality. Chl*a* concentrations of the 200 µL biomass from each filter was extracted via ethanolic extraction and quantified spectrophotometrically according to Ritchie [33]. A one-way ANOVA followed by a Tukey’s multiple comparison test was performed in R (Version 1.2) implemented in RStudio (RStudio Team 2020) to find the statistical significance of Y(II) means (α = 0.05). In both experiments, the average Y(II) values of three experimental replicates per strain were expressed against the two average control values; as both culture and empirical conditions were standardized, it was possible to compare the strain responses to one another directly in an overlaid manner.

### 2.3. Qualitative and Quantitative Osmolyte Analysis

Algal biomass for high-performance liquid chromatography (HPLC) analysis was cultured in two groups. The first cohort was grown in 150 mL liquid 3N BBM+V medium (as in Section 2) enriched with additional 10 mL Provasoli’s enrichment solution [34] per liter, under standard culture conditions outlined in Section 2. The second cohort grew in 150 mL of the same medium at 30 S_A_, again under standard culture conditions. Medium was refreshed weekly by decanting 100 mL from cultivation flasks of followed by replacement with 100 mL fresh medium, in order accelerate biomass accumulation. A single replicate was taken per treatment per strain, due to biomass limitations.

Processing of algal biomass for HPLC was done according to a previously published protocol [8]. In brief, HPLC analysis was done on an Agilent 1260 system (Agilent Technologies, Inc., Santa Clara, CA, USA) equipped with a differential refractive index detector. A Phenomenex REZEX ROA-Organic Acid resin-based column with a Phenomenex Carbo-H+ guard cartridge (Phenomenex, Torrance, CA, USA) was used to separate the solutes. The mobile phase consisted of 5 mM H_2_SO_4_ at a flow rate of 0.4 mL/min at 70 °C.

Solutions comprising 5 mM of sorbitol and sucrose standards were run and quantified by peak areas and retention times. Peak areas in the chromatograms were integrated and correlated to a ten-point calibration curve for each substance detected; resulting concentrations were expressed as µmol g^−1^ algal dry weight (DW). The limit of detection (LoD) was 0.08 mM and 0.09 mM for sorbitol and sucrose, respectively; the limit of quantitation was (LoQ) 0.24 mM and 0.28 mM for sorbitol and sucrose, respectively.

## 3. Results

### 3.1. Morphological Changes under Salt Stress

Cells in standard medium without salt showed no signs of osmotic stress over time, whereas cells that grew in saline medium at both concentrations had clear signs of osmotic stress, evidenced by bleaching, formation of vesicles, reduction in biomass, cell size reduction, and thickened cell walls (Figure 1). The severity of osmotic stress was correlated with increasing salinity. Already at 30 S_A_, there was visible inhibition of growth in strains *Deuterostichococcus tetrallantoides* ASIB-IB-37, *Stichococcus bacillaris* CCAP 379 1/A, *Tetratostichococcus jenerensis* J1302, and *Desmococcus olivaceus* SAG 1.92. Halotolerant strains *Pseudostichococcus monallantoides* SAG 380-1 (Figure 1E) showed only minor physical changes at the cellular level, where cells became smaller compared to growth in 0 S_A_ medium.

Figure 2 gives an overview of the vitality of the strains in the tested media at the end of eight weeks of growth in tabular format; Figure 3 show the macro-scale differences between strains under the same conditions. Most cultures at salinities of 30 S_A_ and above showed markedly reduced biomass, indicative of slowed growth or cell death visible on the macro scale. Growth effectively stopped at 60 S_A_ for the majority of the strains (nine out of 12). Only three strains—SAG 1.92 *Desmococcus olivaceus*, SAG 380-1 *Pseudostichococcus monallantoides*, and *Tritostichococcus solitus* SAG 2406—survived after extended exposure to 90 S_A_. Halotolerant strains, such as SAG 2406 (Figure 3D), maintained more biomass in higher salinities compared to halosensitive strains, such as J1302. There was also a difference in how the biomass was distributed. Certain strains, such as *Deuterostichococcus tetrallantoideus* ASIB-IB-37 (Figure 3C), showed homogenous biomass, whereas others formed a condensed clump under salinity stress (Figure 3A,D). This was not correlated with the morphology of the individual strains, as both filamentous and solitary-form strains could clump or spread, nor with the halotolerance.

### 3.2. Low-Molecular-Weight Carbohydrate Analysis

Some algae including *Stichococcus* are known to produce osmolytes to withstand desiccation stress. The identification and quantification of these low-molecular-weight carbohydrates was verified through HPLC. Due to extremely slow growth in saline medium, it was not possible to obtain sufficient extract for *Desmococcus olivaceus* SAG 1.92. The major compounds corresponded to sorbitol and sucrose, with retention times at 17.1 min and 15.1/16.3 min, respectively (Figure 4; due to hydrolysis by the acidic eluent, sucrose presented a double peak of fructose and glucose).

The typical osmolyte production pattern of a *Stichococcus*-like strain consisted of sorbitol and sucrose in standard (non-saline) medium (Figure 5). The concentration of sorbitol increased in saline growth in all but four strains (SAG 1.92, *Deuterostichococcus marinus* J1303, SAG 11.88 *Diplosphaera epiphytica*, *Pseudostichococcus* sequoieti LB 1820) (Appendix A). Of these four, SAG 11.88 had the largest decrease (176 µmol g^−1^ DW) both in terms of absolute value and percent. The increase in sorbitol after salinity stress of the remaining eight strains varied from ~−85% to 218%, with a median increase of ~29%, corresponding to a median value of 41.5 µmol g^−1^ DW. Strain *Deuterostichococcus tetrallantoides* ASIB-IB-37 showed the highest percent increase in sorbitol (~217%), and strain SAG 380-1 the greatest absolute increase (151 µmol g^−1^ DW).

Four strains (J1303, ASIB-IB-37, SAG 380-1, J1302) had a slightly higher sucrose concentration after the salinity stress, but concentrations generally decreased (median percent change −36%, absolute change −20 µmol g^−1^ DW). The concentration of both osmolytes decreased in four strains (SAG 1.92, SAG 11.88, *Pseudostichococcus undulatus* CALU-1142, CCAP 379/1A *Stichococcus bacillaris*).

### 3.3. Dehydration and Rehydration

To follow the desiccation effect on the cultures, the cells were regularly measured with the PAM using the photosynthetic yield Y(II) as an indicator for the vitality of the algae. The initial Y(II) value of ca. 0.6 is typical of healthy algal strains kept under culture conditions and is not indicative of photoinhibition [35]. The strains had very similar dehydration response patterns (Figure 6), where Y(II) was constant until the ~300 min in a desiccating atmosphere, where some strains’ fluorescence yields began to decline. The majority of the strains dropped precipitously to less than 40% of control values starting at 300 min, and all reached zero by 420 min.

After the rewetting, the algae showed far more variation during the recovery phase than in the dehydration phase. All strains showed a recovery response, with most strains reaching ~30–70% of control values (Appendix A details the average recovery values in both conditions). There were no strains that were functionally dead post-recovery. During the initial two hours of recovery, most strains had low or zero Y(II), but some strains had steadily increasing yield (*Stichococcus bacillaris* SAG 56.91, *Protostichococcus edaphicus* SAG 2481, J1302). SAG 56.91 had an exceptionally high average recovery value, approaching 80% of the control value at the end of the 24 h rehydration period. There were no statistically significant differences between the genera *Stichococcus*, *Protostichococcus*, *Deuterostichococcus*, *Tritostichococcus*, *Tetratostichococcus*, and *Pseudostichococcus* representative strains in response (*p* = 0.283 dehydration; 0.0715 recovery). The highest Y(II) reached by the control was 0.631, on average 0.528, across all control replicates over the duration of 420 min. Appendix A gives final Chl*a* concentrations of the respective filter biomasses.

## 4. Discussion

The *Stichococcus*-like algae are euryhaline and desiccation tolerant but certainly not adapted to marine or hypersaline environments; reports of *Stichococcus*-like algae in marine environments are rare and, as of currently, taxonomically dubious [25,36,37].

### 4.1. Growth in Response to Saline Stress

For this study, we defined halotolerant strains as those that were visually vital (indicated by the green appearance of the culture) and continued to grow at 45 S_A_ and beyond, 1.5 times the average salinity of seawater. Strains that can grow despite saline stress at 30 S_A_ can be considered moderately halotolerant, since this equals marine conditions, and strains that showed heavy signs of stress and cease growth at 30 S_A_ were halosensitive. The growth experiment that 90 S_A_ is the limit of growth for most strains of *Stichococcus*-like algae. Previous work [8] defined a much lower range for good growth of *Stichococcus* sp. (SAG 2060, now *Deuterostichococcus epilithicus*) between 2 and 17 S_A_ but still measured growth at 65 SA. An earlier study [26] found that growth in the strain *Stichococcus bacillaris* CCAP 379/5 ceased at a salinity of 108 S_A_ (strain assignment is not yet updated according to recent revision). Pröschold and Darienko [24] cultivated strains from the *Stichococcus*-like group in marine medium (SWES, 30 S_A_), and all strains were able to survive, despite salinity-induced morphological changes.

Our results confirm that exposure to 30 S_A_ and higher diminishes growth capacity (Figure 2 and Figure 3) for four of the tested strains. Three strains, *Desmococcus olivaceus* SAG 1.92, *Pseudostichococcus monallantoides* SAG 380-1, and *Tritostichococcus solitus* SAG 2406, showed even the potential for growth beyond 90 S_A_ (Figure 2) and are, thus, strongly halotolerant. The four strains *Deuterostichococcus marinus* J1303, *Deuterostichococcus tetrallantoides* ASIB-IB-37, *Pseudostichococcus sequoieti* LB 1820, and *Stichococcus bacillaris* SAG 56.91 are moderately halotolerant and exhibited the beginning of culture collapse at starting at 75 SA. The remaining strains were dead or dying already at 60 S_A_. The clearest signs of stress were chloroplast bleaching, cellular conformation change, and the formation starch and fat droplets and plasmolysis (Figure 1). Plasmolysis was visible in both halosensitive and halotolerant strains at 90 S_A_ and was present in all strains at 30 S_A_ except for in SAG 380-1, LB 1820, and SAG 2460.

Although *Stichococcus*-like algae are commonly aeroterrestrial and found in non-saline habitats, these taxa have been found in marine-adjacent environments, such as sand dune biocrusts affected by ocean spray [38]. Additionally, in the case of the genus *Pseudostichococcus*, these taxa have even been found in manmade habitats such as the highly saline environments of potash-tailings piles [39], which speaks for its halotolerance. Further, *P. monallantoides* SAG 380-1 is one of the few *Stichococcus*-like strains that was isolated from marine habitat (Table 1). It is possible that the genus *Pseudostichococcus* as a whole is more halotolerant than others within the *Stichococcus*-like clade. The hitherto monotypic genus *Pseudostichococcus* has been revised to also include the strains *Pseudostichococcus*
*sequoieti* LB 1820 and *Pseudostichococcus undulatus* CALU-1142 used in this study, and that the latter strains belong to a second, possibly not halotolerant clade, within *Pseudostichococcus* [40]. Further studies are needed to delineate the ecological traits of *Pseudostichococcus* from those of the other *Stichococcus*-like strains.

### 4.2. Compatible Solute Production under Osmotic Stress

Here, we quantified the main osmoprotectant compounds of the *Stichococcus*-like group to link the quantitative change to halotolerance. It was expected that there would be a global increase in both sorbitol (main osmolyte) and sucrose concentrations in saline growth versus standard growth.

HPLC analysis showed increased sorbitol production and lowered sucrose production under saline conditions in most of the strains studied (Figure 5). Despite both sorbitol and sucrose being parallel and direct byproducts of photosynthesis [41], which is depressed under abiotic stress [42,43], there were reduced sucrose levels but generally increased sorbitol levels at 30 S_A_. Nevertheless, the total concentration of osmolytes increased (Appendix A), and the ratio of sorbitol: sucrose changed from approximately 2:1 to 5:1 at 30 S_A_. This prioritization points to the critical role of sorbitol in osmotic stress tolerance.

Four strains (*Desmococcus olivaceus* SAG 1.92, *Diplosphaera chodatii* SAG 11.88, *Pseudostichococcus undulatus* CALU-1142, *Stichococcus bacillaris* CCAP 379/1A) showed major decreases in total osmolyte concentrations following growth in 30 S_A_ compared to 0 S_A_ (Appendix A); only SAG 1.92 was halotolerant. The three remaining strains were halosensitive, and it is possible that saline stress overwhelmed the photosynthesis mechanisms necessary for osmolyte production [42], since there was evidence of bleaching for these three strains at 30 S_A_. The persistent growth of SAG 1.92 under extremely high salinities suggests that there may be additional osmoprotection mechanisms at work that circumvent the polyol protection mechanism, such as Na^+^/H^+^ antiporter gene upregulation, as is the case in cyanobacteria [44], or the presence of Na^+^-ATPases [45,46]. In this case, additional protection may come from the production of a very thick mucilage layer, which the other strains lacked. This layer allows cluster formation and reduces the surface area exposed to medium and may contribute to its survival at 90 S_A_ despite producing very little osmolytes.

The data from this study substantiate the contribution of polyols and sucrose to euryhalinity in aeroterrestrial algae. Polyols acts primarily as an osmolyte due to its high solubility in water and by being biological inert [1], whereas sucrose is a transport and storage molecule [41]; it chiefly protects against desiccation by forming glass structures [47,48]. Based on salt stress data from various studies, polyols seem to contribute more to osmotic stress tolerance compared to mono- or disaccharides. In the stenohaline genus *Klebsormidium*, the main osmolytes are sucrose and raffinose, not polyols, with only very trace amounts of trehalose [49,50], befitting an alga that effectively stop growing at 30 S_A_ [51]. Fujii [14] and Kremer [52] showed that *Heterococcus* was also euryhaline and produced both mannitol and glucose, whose growth stopped at ~45 S_A_. In extremely halotolerant algae, polyols provide insufficient osmotic protection. The alga *Dunaliella salina*, found in extremely saline habitats, produces compatible solutes characteristic of very haloterant algae, such as glycerol and glycine betaine instead of polyols [53,54].

There was no strain-specific relationship between a total osmolyte concentration increase and halotolerance within the *Stichococcus*-like strains. In addition, it was not possible to conclude unequivocally that higher polyol production, either in total or of specific osmolytes, equals higher halotolerance. For example, the most halotolerant strains SAG 2406 and SAG 380-1 had relatively low total osmolyte concentrations, but with a clear sorbitol increase (Appendix A). ASIB-IB-37 had lower total osmolyte concentrations than CCAP 379/1A but was more halotolerant in culture (Figure 2 and Figure 5). This nonetheless advances the extensive work previously undertaken in *Stichococcus* [8,15,20,28,29] but indicates the need for more study of the precise saline stress adaptive mechanisms beyond polyol production within the *Stichococcus*-like genera.

### 4.3. Dehydration and Rehydration

In addition to salinity-induced “physiological drought” [55], dehydration is the second water-related challenge facing aeroterrestrial algae. Since the *Stichococcus*-like clade is not primarily marine, it was assumed that they would be better adapted to desiccation than salt stress.

Dehydration resulting from a lack of water in the environment has different consequences for the cell than osmotic stress, although they are related. Rehydration is usually sufficient to reverse damage to the photosynthetic apparatus, but this is strongly contingent on the extent and length of the desiccation period [56,57]. During the desiccation experiment in this study, algal suspensions were subjected to a long dehydration period (7 h) under very low RH (~10%), comparable to desert-like dryness [57].

Generally, strains that responded faster to desiccation, defined as having a shorter time until zero Y(II), had a lower recovery rate. The high variability of the final Y(II) values suggests that the strains’ responses were sensitive to small changes in experimental preparation, of which the most important were biomass density and pipetting technique, since self-shading would protect from desiccation. Individual strain replicates achieved recovery Y(II) of ~90–95%, but this was not associated with one particular run and cannot be attributed to one consistent mechanical error.

All strains reached, per strain average, at least ~40% of control yield value in the recovery experiment, with the exception of CALU-1142 (~30%). This is exceptionally high as none of the *Stichococcus*-like organisms studied were isolated from deserts; all are temperate or tropical aeroterrestrial, and three strains are even aquatic in origin (SAG 380-1, SAG 56.91, SAG 2406). For perspective, true desert algae are expected to be even more desiccation resistant and able to recovery to higher values after weeks of desiccation [57]; desert green algal taxa outperform aquatic taxa in long-term desiccation and recovery, needing on average 900 min to dehydrate fully, compared to the 420 min from this experiment. Karsten [32] demonstrated in a similar experimental setup that *Klebsormidium* strains from African deserts can recover fully after 20–24 h, with little indication of lasting damage to the photosynthetic system. Other studies in Trebouxiophyceaen algae present on tree bark showed that they, as one biofilm unit, are able to survive and recover after being exposed to extended drought periods of up to 80 days [35]. This was reiterated in [58], where a *Diplosphaera chodatii* isolate collected from tree bark reached nearly 100% initial Y(II) after being exposed to water for post-dehydration, albeit after a far shorter desiccation period (45 min at 10% RH).

At 120 min, most strains reached a limit of ~20% initial Y(II) values. It was not until beyond this time that there was an appreciable uptick overall. Medwed [58] and Karsten [32] showed recovery that reached a plateau at ~190 and ~1000 min, respectively, so it is possible that the *Stichococcus*-like strains needed slightly longer to reach peak recovery.

In contrast to the average recovery value of ~40%, a maximum recovery potential of ca. 80–100% of control values was evident in all the strains, even accounting for high variability (Appendix A). This was the highest value reached by an individual replicate per strain. Still, because all strains were maintained in liquid medium, it is unclear how important environmental origin is, if strains have been acclimated to liquid medium over a period of several months or even years. Morphology seemed to play a limited role in performance here, as most strains were single-celled and little self-shading was possible. Weakly filamentous strains, such as CCAP 379 1/A and SAG 56.91, are easily fragmented by pipetting and acted equally single-celled. Interestingly, there was little correlation between the Chl*a* content of the cell suspensions (biomass), resistance to desiccation, and subsequent recovery potential (Appendix A). The strains with the lowest Chl*a* concentrations (SAG 2481, SAG 406, SAG 380-1) did not show differential responses compared to the strains with higher Chl*a* concentrations. This was also the case with positive outliers, where higher Chl*a* concentration did not result in higher recovery yields.

The data from the salinity stress and desiccation experiments suggest that mechanisms behind ionic and desiccation stress tolerance are independent of one another. Two of the most halotolerant strains (*Tritostichococcus solitus* SAG 2460 and SAG 380-1 *Pseudostichococcus monallantoides*) recovered well after rehydration but not faster than the halosensitive strains. The recovery of SAG 380-1 is comparable to that of CCAP 379/1A *S. bacillaris* and, even, ASIB-IB-37 *Deuterostichococcus marinus*, which had the least total osmolyte production. Despite this, the desiccation tolerance coupled with the ability to survive extended periods of time in saline environments may contribute to the widely known ecological niche of the *Stichococcus*-like genera.

Nevertheless, the experimental tolerance data collected here are to be interpreted as physiological possibilities, not certainties. Most of the strains used have spent extensive time in storage, sometimes decades, within culture collections; it is possible that genomic changes have occurred, when compared to wild-type specimens [59]. Furthermore, while cryopreservation of stock cultures is recommended to minimize changes, it may also lead to an artificial genotypic selection for cryosurvivability [60]. A cause for optimism is that resistance to genetic changes may vary from alga to alga. *Chlorella vulgaris* strain 211-11b is the oldest isolated algal strain in the world, and no differences between it and its many isolates have been found [61]. This is likely the case within the *Stichococcus*-like strains, since the *Deuterostichococcus* sister strains to J 1303, SAG 2139 (MT078164) and UTEX 1706 (DQ27546) have identical sequences apart from length.

For taxonomists, it is important to refer to the oldest described species for taxonomical revision—in the case of microalgae, it mostly relies on culture collections. As this study was to extend the deep taxonomical revision of *Stichococcus*-like species, we decided to use cultured strains (most of them designated as epitypes) for this experimental setup rather than working with environmental specimens. Further studies with environmental samples of the *Stichococcus*-like strains are needed to determine the effects of saline stress and desiccation tolerance in the ecological context.

## 5. Conclusions

The patterns observed in the growth experiments under salinity stress, osmolyte analysis, and desiccation stress experiment indicate that the *Stichococcus*-like strains are euryhaline but better adapted to dehydration than osmotic stress. Strains that withstood salt stress did not necessarily recover better in the desiccation experiment—neither was osmolyte concentration indicating the halotolerance. It is unknown if this is true on a genus-wide basis, since most genera had only one representative taxon strain, but the existing results show no phylogenetic correlation in terms of halo- and desiccation tolerance, with the exception of the very halotolerant genus *Pseudostichococcus*. If this is true, then *Stichococcus* and the newly erected genera *Protostichococcus*, *Deuterostichococcus*, *Tritostichococcus*, and *Tetratostichococcus* may occupy a similar niche as aeroterrestrial algae within the Prasiolaceae. This fits their profile as generalists that are able to colonize arid habitats worldwide as well as those exposed to some saline stress, such as in biocrusts on sand dunes [38], deserts [2], and potash piling heaps’ surroundings [27,39]. However, the performance of cultures should ultimately be compared to that of environmental samples to provide information on their ecology.

Holzinger [62] used transcriptomics to elucidate the molecular mechanisms behind desiccation tolerance of *Klebsormidium*—it would be useful to see if analogous transcripts exist in *Stichococcus*-like organisms, as this could also ground these terrestrial Chlorophyta along the timeline of terrestrial habitat colonization.

## Figures and Tables

**Figure 1 microorganisms-09-01816-f001:**
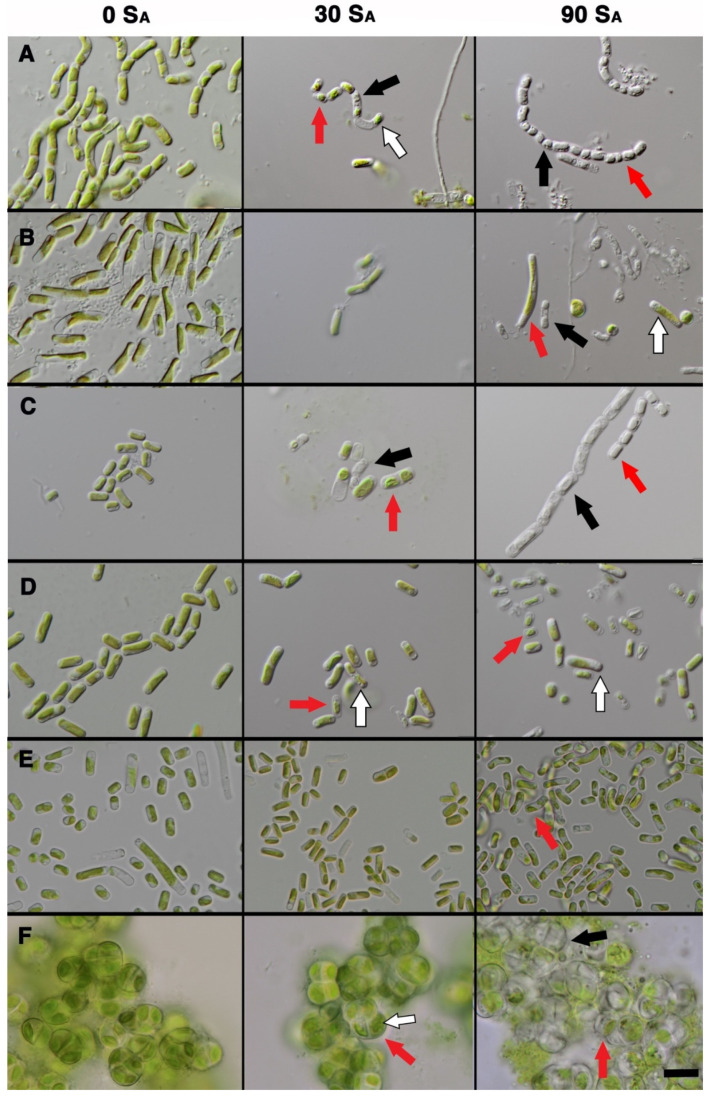
Morphological responses of selected strains in 0, 30, and 90 S_A_ modified 3N BBM+V media after 8 weeks. Scale bar = 5 µm for all panels. (**A**) *Deuterostichococcus tetrallantoideus* ASIB-IB-37. (**B**) *Pseudostichococcus sequoieti* LB 1820. (**C**) *Stichococcus bacillaris* CCAP 379/1A. (**D**) *Tetratostichococcus jenerensis* J1302. (**E**) *Pseudostichococcus monallantoides* SAG 380-1. (**F**) *Desmococcus olivaceus* SAG 1.92. Red arrows indicate plasmolysis, black arrows chloroplast bleaching, and white arrows the accumulation of storage vacuoles. Most strains in the cohort were still able to thrive at 30 S_A_, but in halosensitive strains, there was already some bleaching and cell configuration changes present.

**Figure 2 microorganisms-09-01816-f002:**
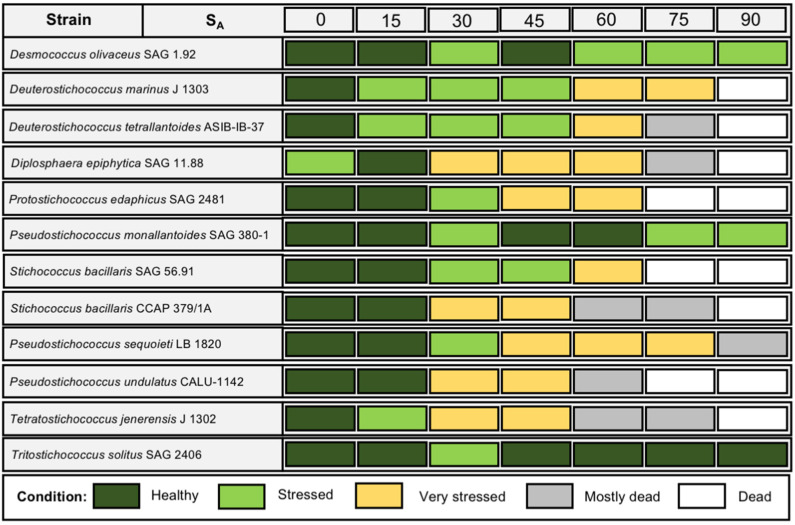
Tabular results of strain survival at 0–90 S_A_; individual cultures of strains were evaluated after 8 weeks of growth in the respective media. The condition colors correspond approximately to culture color and health in the test tubes.

**Figure 3 microorganisms-09-01816-f003:**
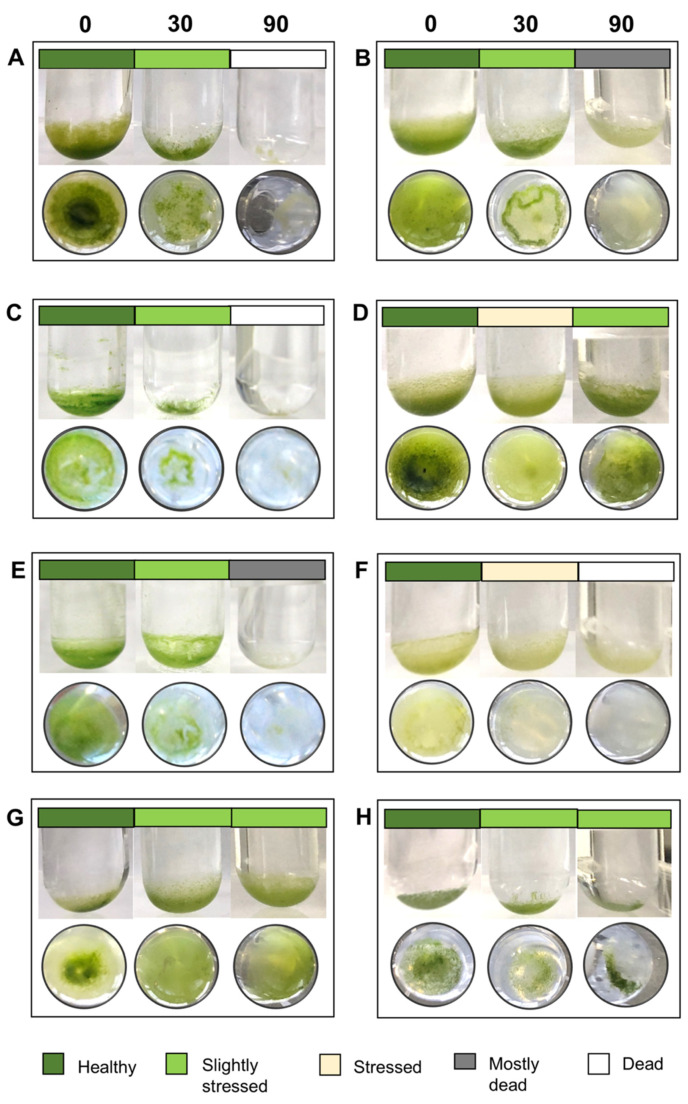
(**A**) SAG 56.91 Stichococcus bacillaris; (**B**) Protostichococcus edaphicus SAG 2481; (**C**) Deuterostichococcus marinus ASIB-IB-37; (**D**) Tritostichococcus solitus SAG 2406; (**E**) Tetratostichococcus jenerensis J1302; (**F**) Diplosphaera epiphytica SAG 11.88; (**G**) Pseudostichococcus monallantoides SAG 380-1; (**H**) Desmococcus olivaceus SAG 1.92. Clumping was observed in (**A**–**C**).

**Figure 4 microorganisms-09-01816-f004:**
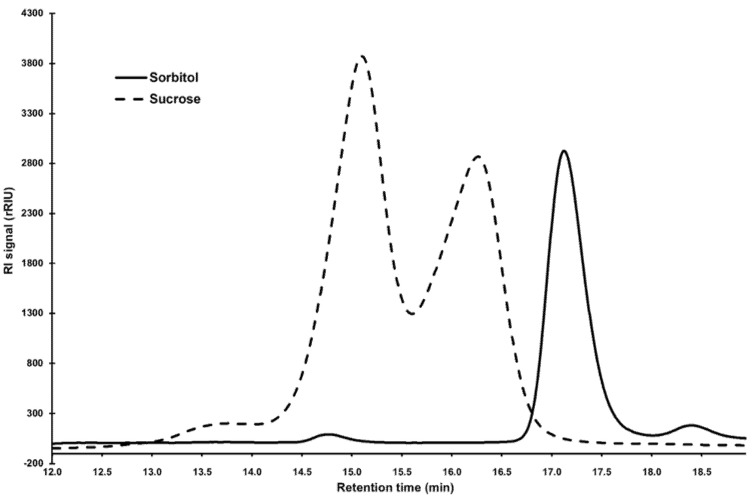
HPLC chromatograms of cellular extracts of pure standards of sorbitol and sucrose, respectively. Sorbitol shows a peak at 17.1 min; sucrose at 15.1 (glucose) and 16.3 min (fructose). RI signal = Refractive Index signal; rRIU = relative Refractive Index Units.

**Figure 5 microorganisms-09-01816-f005:**
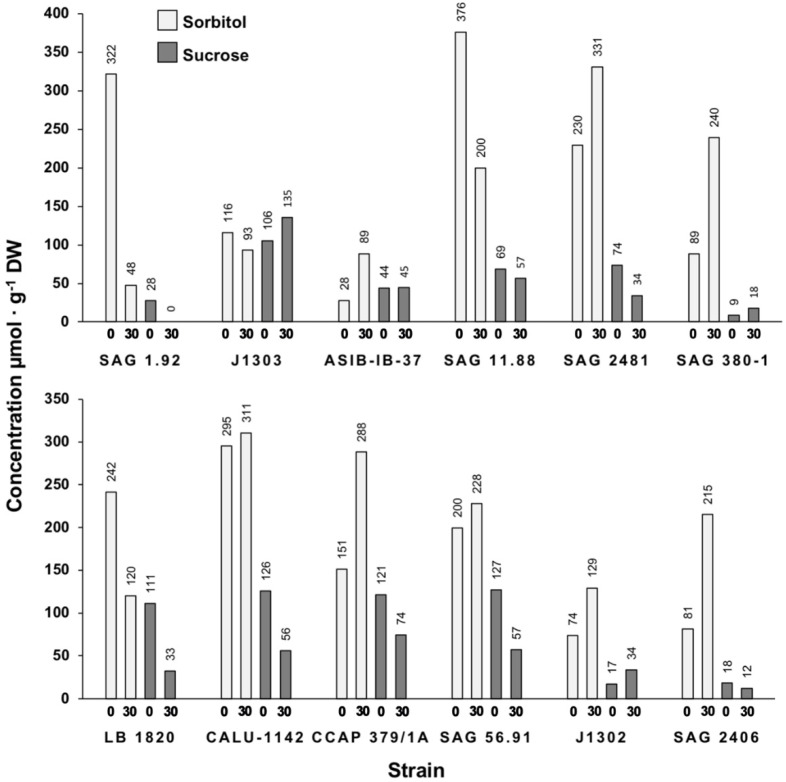
Proportions of the major osmolytes present in each strain during growth at 0 and 30 S_A_. Concentrations are expressed as µmol g^−1^ dry weight (DW).

**Figure 6 microorganisms-09-01816-f006:**
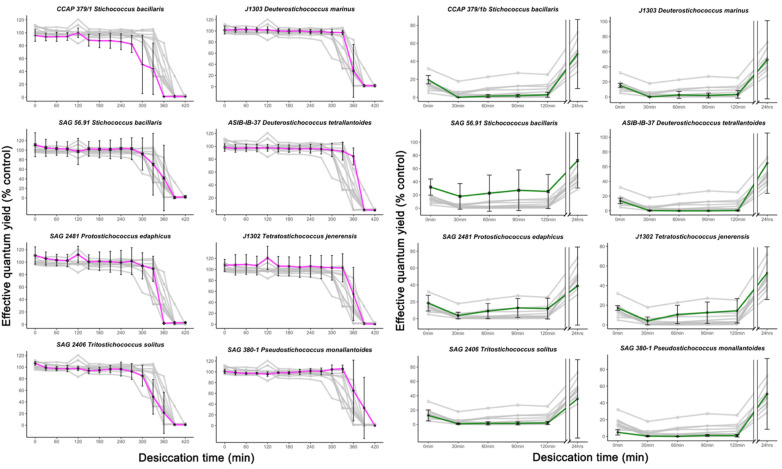
Concentrations are expressed as µmol g^−1^ dry weight (DW). Comparison of Y(II) after desiccation and rehydration in selected *Stichococcus*-like strains. The strains are representative of the major lineages within the clade. Magenta lines show average effective quantum yield (Y(II)) per strain during dehydration, while green lines show the average Y(II) in the recovery phase; grey lines in the background represent the other strains for comparison. Error bars represent standard deviation (*n* = 3).

**Table 1 microorganisms-09-01816-t001:** The strains used in this study and their culture ID, localities, and collector information. Equivalent strains from different culture collections are marked with a “^†^”; strains marked with “*” are authentic strains. All strains except SAG 1.92 were used in the dehydration and recovery experiment; all strains were used in the salt growth experiment.

Strain ID	Species Assignment	Locality and Habitat	Collector/Isolator
SAG 1.92 *	*Desmocococcus olivaceus*	Vienna, Austria; subaerial	W. Vischer, before 1960
J 1303 *, ^†^ SAG 2139 *	*Deuterostichococcus marinus*	Dauphin Island, Alabama, USA; soil	T.R. Deason, 1969
ASIB-IB-37 *	*Deuterostichococcus tetrallantoideus*	Weißkugel Peak, Ötztal Valley, Austria; soil	H. Reisigl, 1964
SAG 11.88 *	*Diplosphaera epiphytica*	Waweira Scenic Reserve, New Zealand; lichen phycobiont	E. Tschermak-Woess, 1984
SAG 2481 *	*Protostichococcus edaphicus*	Swabian Alb, Germany; forest soil	L. Hodač, 2008
SAG 380-1 *	*Pseudostichoccocus monallantoides*	Germany; marine	L. Moewus, 1951
LB 1820 *	*Pseudostichococcus sequoieti*	USA; redwood forest soil	G. Arce, 1971
CALU-1142 *	*Pseudostichococcus undulatus*	Dolomite Mountains, Italy	G. Vinatzer, 1975
CCAP 379/1A *	*Stichococcus bacillaris*	Likely Switzerland	W. Vischer, before 1936
J 1302 *, ^†^ SAG 2138 *	*Tetratostichococcus jenerensis*	Kampong Kuala Jenera, Kelantan, Malaysia; soil of rainforest tree	J. Neustupa, 2000
SAG 2406 *	*Tritostichococcus solitus*	Northeim, Germany; karstwater stream rock surface	K. Mohr, 2003

## Data Availability

Data are contained within the article.

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
