# Peer review of "The Ecophysiological Performance and Traits of Genera within the Stichococcus-like Clade (Trebouxiophyceae) under Matric and Osmotic Stress"

_microorganisms, 2021, doi:10.3390/microorganisms9091816_

Round 1

Reviewer 1 Report

The work by Tu Van et al. aims at describing the salt- and desiccation stress-tolerance of several strains from the Stichococcus-like clade. Apart from visual observation of the culture growth and condition at different salinities, as well as during drying-rehydration cycle, major carbohydrate osmolyte content and photosynthetic activity (via PAM chlorophyll fluorescence) were monitored. A large body of new interesting data was obtained, including some important observations e.g. of lack of correlation between the observed tolerance and osmolyte level in the cells. The paper has the potential to be published but the manuscript suffers from some flaws which should be dealt with first (please see the list below).

Please explain the non-standard term “matric (stress)” at first occasion so the readers would correctly understand it. How is it different e.g. from desiccation stress?

L15: environ-mental --> environmental

L47: 6/8/2021 11:13:00 AM — ???

L81: what was the final cell density/Chl content after the inoculation?

L96: what was the cell density/Chl content of the cell suspension used?

L125: what is meant by the “media refreshment”? What was the ratio of old/added media volumes?

Fig. 1: the same scale bar of 5 µm is valid for all panels of this figure? Please state this explicitly in the legend.

Fig. 4: trehalose peak is missing? What are the additional peaks obtained for sucrose—glucose and/or fructose? What are the assignments of the peaks?

Fig. 5: are the observed differences statistically different in all cases? Please indicate the (results of) the statistic treatment used.

Fig. 6: it would be enough to show the averages ± SD (?, please indicate if the error bars are SD), the kinetics for the individual replicas make the figure overloaded.

Author Response

Supplementary Table 3 is pasted below - it will be uploaded to the portal later. 

Strain

mg L-1

J1303 Deuterstichococcus marinus

22.27 ± 12.00

ASIB-IB-37 Deuterostichococcus tetrallantoideus

26.69 ± 17.74

SAG 11.88 Diplosphaera epiphytica

28.22 ± 3.640

SAG 2481 Protostichococcus edaphicus

8.793 ± 3.246

SAG 380-1 Pseudostichococcus monallantoides

5.526 ± 1.638

LB 1820 Pseudostichocococcus sequoieti

14.28 ± 9.455

CALU-1142 Pseudostichococcus undulatus

25.63 ± 8.925

CCAP 379/1 Stichococcus bacillaris

25.16 ± 13.21

SAG 56.91 Stichococcus bacillaris

37.51 ± 8.627

J1302 Tetratostichococcus jenerensis

17.65 ± 10.96

SAG 2406 Tritostichococcus­­ solitus

5.480 ± 4.585

Tab. S3. The average Chla concentrations of 200 µL biomass dropped onto the filters of the dehydration experiment, expressed as mg Chla ∙ L-1 (n = 3 per strain). 

Reviewer 2 Report

I don't think the results of this article are very innovative. This experimental design is relatively simple overall. The author aims to characterize the eco-physiological performance of Stichococcus-like strains under osmotic as well as matric stress, pragmatically realized as salt and desiccation tolerance. Only old algal strains were used, but no ecological data were available. I don't think these old algal strains are a good way to identify the ecological niche of the species.

Author Response

Supplementary Table S3 is pasted below. It will be uploaded to the portal afterwards.

Strain

mg L-1

J1303 Deuterstichococcus marinus

22.27 ± 12.00

ASIB-IB-37 Deuterostichococcus tetrallantoideus

26.69 ± 17.74

SAG 11.88 Diplosphaera epiphytica

28.22 ± 3.640

SAG 2481 Protostichococcus edaphicus

8.793 ± 3.246

SAG 380-1 Pseudostichococcus monallantoides

5.526 ± 1.638

LB 1820 Pseudostichocococcus sequoieti

14.28 ± 9.455

CALU-1142 Pseudostichococcus undulatus

25.63 ± 8.925

CCAP 379/1 Stichococcus bacillaris

25.16 ± 13.21

SAG 56.91 Stichococcus bacillaris

37.51 ± 8.627

J1302 Tetratostichococcus jenerensis

17.65 ± 10.96

SAG 2406 Tritostichococcus­­ solitus

5.480 ± 4.585

Tab. S3. The average Chla concentrations of 200 µL biomass dropped onto the filters of the dehydration experiment, expressed as mg Chla ∙ L-1 (n = 3 per strain). 

Round 2

Reviewer 1 Report

The authors have addressed most of my criticism. However, I still do not understand why the standard sample of sucrose gives two peaks of glucose and fructose instead of a single peak. Was it accidentally hydrolysed?

Author Response

Reply to Reviewer 1, Round 2

1) The authors have addressed most of my criticism. However, I still do not understand why the standard sample of sucrose gives two peaks of glucose and fructose instead of a single peak. Was it accidentally hydrolysed?

-----

The eluent for HPLC was 5 mM H2SO4, which did cause acid hydrolysis that led to two peaks for sucrose (L232-233; L298-299).

Reviewer 2 Report

"In the second case, we trust that culture collections have maintained their strains in a manner that minimizes morphological and metabolic changes; there are no changes to the genetic code to be expected, else that would undermine the utility of culture collections."

-- According to recent research, this response is not acceptable. Please see Yamagishi et al. (2020, https://doi.org/10.1371/journal.pone.0241889) You should provide evidence or reference that the genetic code will not change.  
